# Improving malaria case management with artemisinin-based combination therapies and malaria rapid diagnostic tests in private medicine retail outlets in sub-Saharan Africa: A systematic review

**Catherine Goodman**[1]*, **Sarah Tougher**[1], **Terrissa Jing Shang**[1], **Theodoor Visser**[2]

**1** Department for Global Health and Development, London School of Hygiene and Tropical Medicine, London, United Kingdom, **2** Clinton Health Access Initiative, Inc. (CHAI), Global Malaria, Boston, Massachusetts, United States of America

* catherine.goodman@lshtm.ac.uk

## Abstract

Private medicine retailers (PMRs) such as pharmacies and drug stores account for a substantial share of treatment-seeking for fever and malaria, but there are widespread concerns about quality of care, including inadequate access to malaria rapid diagnostic tests (RDTs) and artemisinin-based combination therapies (ACTs). This review synthesizes evidence on the effectiveness of interventions to improve malaria case management in PMRs in sub-Saharan Africa (PROSPERO #2021:CRD42021253564). We included quantitative studies evaluating interventions supporting RDT and/or ACT sales by PMR staff, with a historical or contemporaneous control group, and outcomes related to care received. We searched Medline Ovid, Embase Ovid, Global Health Ovid, Econlit Ovid and the Cochrane Library; unpublished studies were identified by contacting key informants. We conducted a narrative synthesis by intervention category. We included 41 papers, relating to 34 studies. There was strong evidence that small and large-scale ACT subsidy programmes (without RDTs) increased the market share of quality-assured ACT in PMRs, including among rural and poorer groups, with increases of over 30 percentage points in most settings. Interventions to introduce or enhance RDT use in PMRs led to RDT uptake among febrile clients of over two-thirds and dispensing according to RDT result of over three quarters, though some studies had much poorer results. Introducing Integrated Community Case Management (iCCM) was also effective in improving malaria case management. However, there were no eligible studies on RDT or iCCM implementation at large scale. There was limited evidence that PMR accreditation (without RDTs) increased ACT uptake. Key evidence gaps include evaluations of RDTs and iCCM at large scale, evaluations of interventions including use of digital technologies, and robust studies of accreditation and other broader PMR interventions.

**Data Availability Statement:** All relevant data are within the manuscript and its Supporting information files.

**Funding:** This review was funded by the Global Malaria Programme of the World Health Organisation https://www.who.int/teams/global-malaria-programme (award received by CG). The funder provided comments on the study protocol and manuscript, and gave permission for submission for publication.

**Competing interests:** The authors have declared that no competing interests exist.

# Introduction

Malaria remains a leading cause of the burden of disease in Africa, accounting for an estimated 228 million cases and 602 000 deaths in 2020, around 95% of global malaria cases and deaths [1]. The past 10–15 years have witnessed considerable progress in case management of malaria, particularly through the introduction of highly effective artemisinin-based combination therapies (ACTs) as first-line treatment, and the expansion of parasitological diagnosis. Since 2010 the World Health Organisation (WHO) has recommended parasitological diagnosis for every suspected malaria case, as clinical malaria is indistinguishable from the early stages of many other diseases, meaning that reliance on clinical diagnosis alone leads to a substantial proportion of patients being treated with antimalarials when their illness has a non-malarial cause [2]. The feasibility of parasitological diagnosis has also been substantially enhanced by the introduction of malaria rapid diagnostic tests (RDTs) which are quick (<20 min), accurate, simple to use, and relatively inexpensive [3].

While ACTs and RDTs have been widely adopted in public healthcare facilities across Africa [4], coverage remains inadequate in the private sector, which accounts for a high proportion of treatment seeking for malaria and fever. Among febrile children under 5 for whom care was sought, 30.8% sought care in the private sector based on household surveys conducted between 2015–19 [1], and the percent is likely higher in older age groups. Private sector use is particularly high in some of the highest malaria burden countries in sub-Saharan Africa, accounting for over 50% of treatment seeking for sick children in Nigeria, Uganda, and Tanzania [5]. The private sector includes private facilities (such as clinics and hospitals) and private medicine retailers (PMRs). PMRs encompass both retail pharmacies that should be staffed by someone with a pharmacy qualification, and drug shops with less qualified staff, that are allowed to sell a more limited range of medicines, variously known as patent medicine vendors, over-the-counter medicine sellers or dépôts de médicaments. In some settings, medicines are also available through general stores, market stalls and hawkers. PMRs account for a substantial share of private sector treatment seeking for fever, accounting for example for over 60% of private sector visits for sick children in Tanzania, the Democratic Republic of Congo (DRC), and Nigeria [5].

There are widespread concerns about the quality of malaria case management received from PMRs. Febrile patients treated through PMRs typically receive no parasitological diagnosis, as microscopy is not feasible in retail settings and RDT availability is very patchy [6]. Although in most countries with high private sector use some ACTs can legally be sold without a prescription in pharmacies and drug stores [7], in practice, patients often purchase less effective antimalarials or no antimalarial treatment [8]. In addition, there are concerns around the quality of ACTs [9], lack of referral to public sector facilities, inappropriate use of antibiotics, and the failure to include malaria cases treated by PMRs in national surveillance data [10].

A range of strategies have been developed to address these concerns. In the pre-ACT era, programmes typically involved some combination of PMR training, job aids, pre-packaging of tablets, and social marketing, with occasional use of accreditation and franchising [11]. With the introduction of ACTs, which were initially substantially more expensive than more commonly used antimalarials such as chloroquine, more radical action was taken to reduce ACT prices in the private sector. ACTs were distributed with a significant subsidy with the intent of reducing retail prices for patients, and thus increasing ACT demand and crowding out other, often less effective, antimalarials. ACT subsidies began through small-scale pilots, but were then adopted on a massive scale under the Affordable Medicine Facility-malaria (AMFm). The AMFm was established by the Global Fund to Fight AIDS, Tuberculosis and Malaria, covering 7 African countries: Ghana, Kenya, Madagascar, Niger, Nigeria, Uganda, and Tanzania from

2010–12 [12, 13]. It subsidised quality-assured ACTs, defined as those pre-qualified by WHO or approved by Stringent Regulatory Authorities. Following this period, the Global Fund continued private sector ACT subsidies through the private sector co-payment mechanism (CPM) in six countries [8]. Unlike the AMFm which had earmarked funding, countries were required to allocate funds to the CPM from their core Global Fund malaria grant.

In parallel, various groups began testing interventions to introduce RDTs in PMRs [14]. Other groups focused more broadly on treatment of childhood illness, adapting Integrated Community Case Management (iCCM) guidelines for the treatment of malaria, pneumonia and diarrhoea for drug stores [15, 16]. A few countries, beginning with Tanzania, implemented large-scale accreditation schemes for PMRs, including sales of subsidised ACTs [17].

More recent years have seen fewer large-scale PMR malaria interventions in sub-Saharan Africa, but there is currently renewed interest in this topic among national and multilateral organizations [18, 19], reflecting the continued high treatment seeking rates in this sector. As no comprehensive, up-to-date systematic review was identified on strategies to improve care of febrile patients by PMR, WHO commissioned this review. The objective was to synthesize the evidence on the effect of interventions relating to malaria case management on improving care received from PMRs in sub-Saharan Africa, based on evaluations with a historical or contemporaneous control group. It updates the review by Visser et al. on RDT introduction in PMRs [14], while also broadening the scope to consider any intervention relevant to malaria case management.

## Methods

### Eligibility criteria

This systematic review followed the 2020 PRISMA-p guidelines and was registered on PROSPERO (2021: CRD42021253564) where the protocol can be accessed.

We included studies conducted in sub-Saharan Africa only, reflecting the high malaria burden of many of these countries. While there is considerable literature on malaria case management in PMRs from outside Africa, particularly from the Greater Mekong Subregion [20], we focus on sub-Saharan Africa given the substantial differences in malaria epidemiology and retail market structure between regions.

We included all published and unpublished randomized and non-randomized controlled trials, pre-post designs with or without a control group, and time-series or repeated measure surveys, that reported on an intervention to support the provision of RDTs and/or ACTs by PMR staff. We included evaluations of both malaria-specific interventions and broader interventions that encompass ACTs or RDTs. Studies that did not report on original research, such as opinion pieces and literature reviews, were excluded, as were reports or conference abstracts with insufficient methodological detail.

We defined PMRs as pharmacies, drug stores and any other for-profit private sector retailer supplying medicines. Studies that only reported on for-profit healthcare facilities such as clinics, health centers, and hospitals, not-for-profit providers or community health workers were excluded. To be eligible, studies must have reported on at least one of our primary outcomes, which comprised any measurement of the effect of an intervention on ACT and RDT uptake, dispensing treatment according to test results, referral on the basis of protocol, patient adherence to the treatment regimen, antibiotic uptake, antimalarial quality, and health outcomes (further details provided below). These outcomes were selected as they relate directly to the care received by users, as opposed to intermediate outcomes such as staff knowledge, product availability or price.

We included studies where data were collected between January 2006 and March 2023. The time frame was chosen because prior to 2006, PMR interventions did not involve ACTs or RDTs, and therefore their findings are expected to be less applicable to the current context.

## Search strategy

A systematic literature search for published studies was conducted on March 10th, 2023, in Medline Ovid, Embase Ovid, Global Health Ovid, Cochrane Library and Econlit Ovid. The search terms were synonyms, related terms and MESH terms for five domains (i) 'malaria and fever' (ii) 'PMRs' (iii) 'diagnosis and treatment' (iv) 'antimalarials' and (v) 'Sub-Saharan Africa' (see full search strategy in S1 Table).

The search terms were strategically combined, and their performance tested by comparing a list of pre-identified eligible papers to the search results. The results were then filtered by the time limit (Jan 2006-March 2023) and language (English and French). The results of the final searches were exported into Covidence, and duplicates removed.

Covidence was used for independent screening by two authors (ST & TJS). Title and abstracts were screened to identify papers that reported on interventions pertaining to ACT and/or RDT within the private sector, with studies excluded that did not meet the study design criteria. For papers meeting the screening criteria, a full-text review was conducted to determine eligibility. Discrepancies were resolved by discussion and consultation with a third author (CG). Reference lists of all included studies and of earlier related literature reviews [11, 14–16, 21–23] were examined to check for additional papers not captured in the search.

Unpublished studies were identified by contacting researchers and practitioners in the field; a full-text review was conducted independently by ST and CG to identify eligible studies.

## Data extraction, outcomes, and quality assessment

We conducted a narrative synthesis rather than meta-analysis because of the high degree of heterogeneity across studies in terms of their study design, outcomes, intervention and context. The studies were classified into four broad intervention categories, and the intervention activities, data collection methods, and primary outcomes extracted for each study.

Eligible studies were tabulated to document the direction, magnitude and significance of effects for each reported primary outcome. The following primary outcomes were extracted and synthesized:

1. **ACT uptake:** The proportion of patients with a history of fever who received an ACT (or a quality-assured ACT) (use), or the proportion of antimalarials dispensed that are ACT (or a quality-assured ACT) (market share).

2. **RDT uptake:** The proportion of patients with a history of fever who received an RDT.

3. **Dispensing treatment according to test results:** Typically, the proportion of patients who received no antimalarial after a negative RDT result and the proportion of patients who received ACT after a positive RDT result (though there was some variation in definitions used).

4. **Referral:** The proportion of patients requiring referral by intervention protocol who were appropriately referred; and the proportion of patients referred who completed the referral.

5. **Patient adherence:** The proportion of patients dispensed ACT who completed the full dose as directed.

6. **Antibiotic uptake:** The proportion of patients with a history of fever that received an antibiotic.

7. **Antimalarial quality:** The proportion of antimalarials purchased or sold that passed medicine quality control tests.

8. **Health outcomes:** Any health-related outcomes including but not limited to mortality, morbidity, recovery rates, further treatment-seeking, and post-treatment anemia.

The methodological quality of each study was assessed using an adjusted Downs and Black quality assessment instrument that was developed to appraise the quality of quantitative studies [24]. After testing various quality assessment tools, Downs and Black was chosen as the primary tool due to its ability to capture the wide range of study designs in this review. Minor adjustments were made to the tool to better assess the range of studies included in this review, including the addition of questions from the Effective Public Health Practice Project quality assessment tool for quantitative studies, Drummond's checklist for assessing economic evaluations, and the Newcastle-Ottawa Scale for assessing the quality of nonrandomised studies in meta-analyses (S2 Table) [25–27]. The quality assessment for each study was conducted by two authors independently. The results were used to inform the narrative synthesis, with any concerns about specific studies noted in the results.

The results were synthesized within each intervention category. Factors such as context, background, the difference in participants, variations in interventions, quality of evidence, and outcomes were considered to understand the differences in direction and size of effect across studies.

## Choice of data source reported

The choice of data source for outcomes such as RDT uptake and medicines dispensed can have an important impact on study findings. Sources may include extracting data from registers or other records that providers keep themselves (provider records), direct observation of provider activities by research team staff present at the PMR, exit interviews with clients leaving a PMR, patient follow-up surveys (where patients are telephoned or visited at home after their PMR visit), and cross-sectional household and outlet surveys. The potential for a Hawthorne effect (where PMRs alter their behaviour because they are being observed) or social desirability bias (where they record or state what they believe are correct behaviours) are likely to be highest with provider records and direct observation, followed by exit interviews and outlet surveys, but lower for patient follow-up surveys or household surveys. However, the potential for recall bias is likely to be highest in household surveys, which may also be more susceptible to confounding by contextual changes in the health system e.g. public sector stockouts. To take account of this in the results we note the data source whenever we describe studies. Where a study has more than one data source for a given indicator we generally present exit interviews (rather than direct observation or household surveys).

## Results

### Study selection

A total of 2598 records were identified through the database searches, plus 4 records identified through other sources (Fig 1). After removing duplicates, 1466 records were screened by title and abstract, and 168 full text records were assessed for eligibility. Of these 41 papers were included in the review, relating to 34 distinct studies.

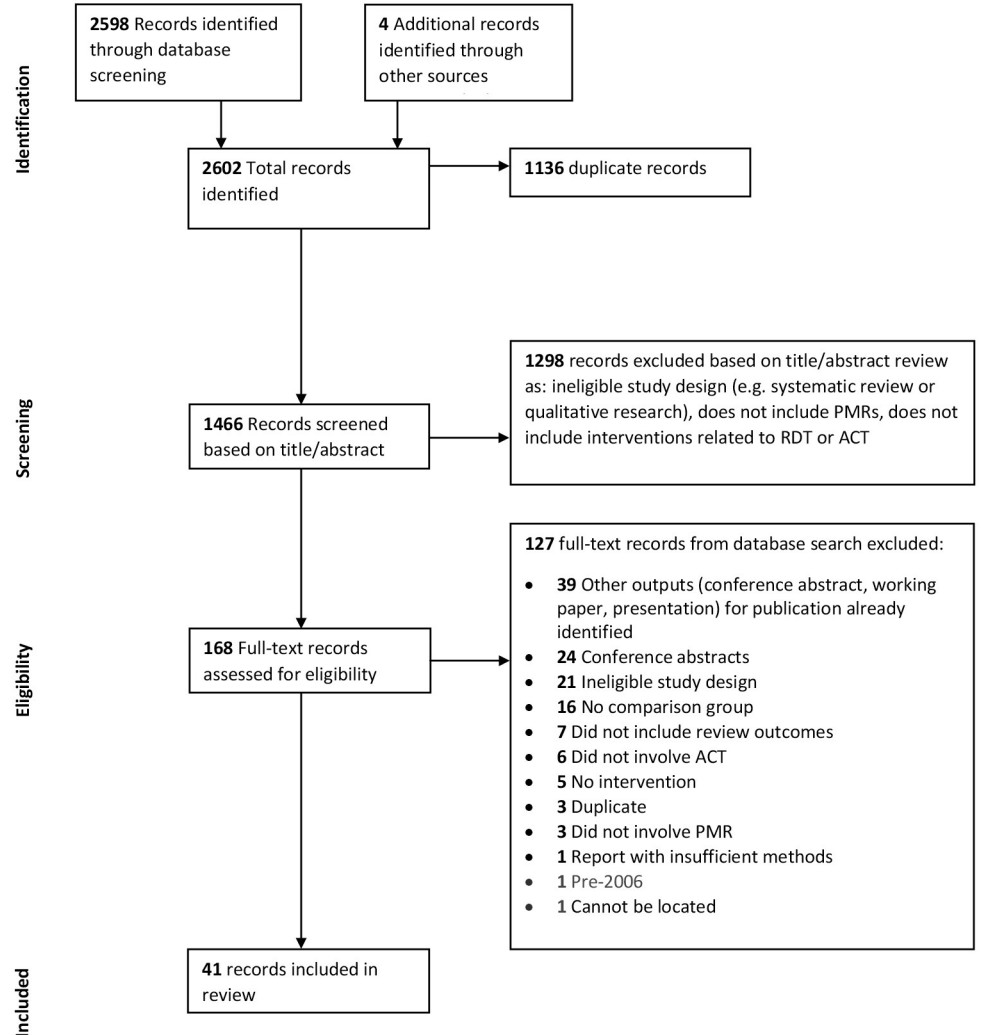

**Fig 1. PRISMA flow diagram.**

The studies were categorised into four groups based on intervention content and breadth: (i) introducing and enhancing ACT use (with no RDTs) (12 studies); (ii) introducing and enhancing RDT use (15 studies); (iii) integrated community case management (iCCM) of malaria, pneumonia and diarrhoea (4 studies); and (iv) broader PMR strategies that included malaria treatment (3 studies).

The characteristics of included studies are summarised in Table 1. Study findings are presented in Tables 2 and 3. As the main outcome indicators are different for interventions with and without RDTs, interventions without RDTs are presented in Table 2 and those with RDTs in Table 3. The Supporting Information provides detailed characteristics of each study (S3 Table), data collection methods (S4a and S4b Table), quality assessment of studies (S5 Table). It also includes heat maps which colour-code the intensity of the intervention and intervention outcomes, in order to explore patterns of results across studies (S6a and S6b Table).

We present the results by each of the four intervention categories.

**Table 1. Summary of study characteristics[1].**

| | Introducing and enhancing ACT use (without diagnostics) n = 12 studies | Introducing and enhancing RDT and ACT use n = 15 studies | Introducing and enhancing iCCM n = 4 studies | Broader private sector strategies including ACT n = 3 studies | Total n = 34 studies |
|---|---|---|---|---|---|
| **Study design:** | | | | | |
| **Cluster randomised control trial** | 2 | 7 | 1 | 1 | 11 |
| **Individually randomised control trial** | 2 | 5 | - | - | 7 |
| **Pre and post without control** | 5 | - | - | - | 5 |
| **Pre and post with control** | 3 | 2 | 2 | - | 7 |
| **Interrupted time series with control** | - | - | 1 | - | 1 |
| **Post-intervention with control** | - | 1 | - | 2 | 3 |
| **Country[2]:** | | | | | |
| **Angola** | 1 | - | - | - | 1 |
| **Ghana** | 3 | 2 | - | - | 5 |
| **Kenya** | 3 | 5 | - | - | 8 |
| **Madagascar** | 2 | - | - | - | 2 |
| **Niger** | 1 | - | - | - | 1 |
| **Nigeria** | 3 | 4 | - | - | 7 |
| **Tanzania** | 5 | 1 | - | 2 | 8 |
| **Uganda** | 6 | 3 | 4 | 1 | 14 |
| **PMR type:** | | | | | |
| **Drug shop** | 4 | 11 | 4 | 2 | 21 |
| **Pharmacy** | 1 | 2 | - | - | 3 |
| **Drug shops and pharmacies** | - | 2 | - | - | 2 |
| **Any PMR** | 7 | - | - | - | 7 |
| **CHWs as retailers** | - | - | - | 1 | 1 |
| **Location:** | | | | | |
| **Rural** | 5 | 9 | 3 | 1 | 18 |
| **Urban (inc. peri-urban)** | - | 2 | - | - | 2 |
| **Rural and Urban (or peri-urban)** | 7 | 4 | 1 | 2 | 14 |

[1] Full details of characteristics of each study are provided in S3 Table

[2] 3 ACT subsidy studies covered multiple countries

## Introducing and enhancing ACT use (without diagnostics)

The 12 studies on introducing or enhancing ACT use all included retail sector ACT subsidies. The findings are presented in 3 groups: evaluations of sub-national ACT subsidies (4 studies), national ACT subsidies (5 studies, of which 3 are multi-country), and interventions to improve adherence to subsidised ACT (3 studies) (Table 2).

**Sub-national ACT subsidy programmes.** Four evaluations were of sub-national ACT subsidies in Kenya, Uganda, Tanzania and Angola [28, 30–32]. All the projects started pre-AMFm, lasting 8 months to 3 years. The interventions were similar, involving an ACT subsidy

**Table 2. Study outcomes—Introducing and enhancing ACT use (without diagnostics), and broader private sector strategies including ACT—% (95% CI or SD) [SE].**

| Study Design | | Outcomes[1] | | | | | | |
|---|---|---|---|---|---|---|---|---|
| First author, Yr published; Country; Data source | Study Arms | ACT uptake | | | Antibiotic uptake | Patient adherence to treatment regimen | Medicine quality | Health outcomes |
| | | % febrile patients receiving ACT | % febrile patients receiving an antimalarial that received an ACT | % of antimalarial sales volumes that were ACTs | % of febrile patients that received an antibiotic | % of patients dispensed ACT who complete the full dose as directed | % of antimalarials purchased or sold that meet quality standards | |
| **1. Introducing and enhancing ACT use (without diagnostics)** | | | | | | | | |
| **1.1 Sub-national ACT subsidy programmes** | | | | | | | | |
| Kangwana 2011, 2013 [28, 29] Kenya<br><br>Household survey; mystery shopper survey (market share only) | Intervention: Subsidised paediatric ACTs | 53.7 (12.3)*[2] | 25.4 (6.9)*[3] | | | 67.0 (8.5)[2] | | |
| | Control: No subsidy | 27.3 (15.2)[2] | 1.8 (1.3)[2, 3] | | | 49.4 (24.8)[2] | | |
| Lussiana 2016 [30] Angola<br><br>Outlet survey | Intervention: Subsidised paediatric ACTs | | | 52.8~[4] | | | | |
| | Baseline: No subsidy | | | 0.0 | | | | |
| Sabot 2009 [31] Tanzania<br><br>Exit interview (use); outlet survey (sales volumes) | Intervention: Subsidised ACTs and RRP | | 70.8~ (<5 years)[5] 31.1~ (>5 years) | 60.3*^[36] | | | | |
| | Intervention: Subsidised ACTs and no RRP | | 40.8~ (<5 years) 48.1~ (>5 years) | | | | | |
| | Control: No subsidy | | 6.3 (<5 years) 0~ (>5 years) | Negligible | | | | |
| Talisuna 2012 [32] Uganda<br><br>Exit interview | Intervention: Subsidised ACTs | 26.2* (23.2–29.2) | 74.0 | | | 82.8~[10] | | |
| | Control: No subsidy | 5.6 (4.0–7.35) | Not stated | | | Not stated | | |
| **1.2 National ACT subsidy programmes** | | | | | | | | |
| ACTwatch 2017; IE Team 2012; Tougher 2012 [12, 33] Ghana<br><br>Outlet survey | Intervention: AMFm | | | 51.8*[7] (47.9–55.7) | | | | |
| | Baseline: No subsidy | | | 6.5[7] (3.1–9.8) | | | | |
| Kenya<br><br>Outlet survey | Intervention: CPM 2014 | | | 48.2*[7,9] | | | | |
| | Intervention: AMFm | | | 61.4*[7] (53.6–69.2) | | | | |
| | Baseline: No subsidy | | | 12.1[7] (6.0–18.2) | | | | |

(*Continued*)

**Table 2.** (Continued)

| Study Design | | Outcomes[1] | | | | | | |
|---|---|---|---|---|---|---|---|---|
| First author, Yr published; Country; Data source | Study Arms | ACT uptake | | | Antibiotic uptake | Patient adherence to treatment regimen | Medicine quality | Health outcomes |
| | | % febrile patients receiving ACT | % febrile patients receiving an antimalarial that received an ACT | % of antimalarial sales volumes that were ACTs | % of febrile patients that received an antibiotic | % of patients dispensed ACT who complete the full dose as directed | % of antimalarials purchased or sold that meet quality standards | |
| **Madagascar** <br><br> **Outlet survey (sales volumes); household survey (use)** | Intervention: CPM 2015 | | | 7.0*[7,8,9] | | | | |
| | Intervention: CPM 2013 | | | 32.0*[7,8,9] | | | | |
| | Intervention: AMFm | 8.4*[10] (4.9–14.3) | 44.5~[10] (28.8–61.4) | 22.0[7] (12.9–31.0) | | | | |
| | Baseline: No subsidy | 3.4[10] (2.0–5.7)[0] | 7.2[10] (4.4–11.5) | 6.8[7] (3.6–10.1) | | | | |
| **Niger** <br> **Outlet survey** | Intervention: AMFm | | | 18.1*[7] (12.6–23.6) | | | | |
| | Baseline: Subsidy | | | 3.7 (1.2–6.2) | | | | |
| **Nigeria** <br><br> **Outlet survey (sales volumes); household survey (use)** | Intervention: CPM 2015 | | | 35.0*[7,8,9] | | | | |
| | Intervention: CPM 2013 | | | 27.0*[7,8,9] | | | | |
| | Intervention: AMFm | 11.1*[10] (9.0–13.6) | 30.3~[10] (25.0–36.1) | 17.8*[7] (14.4–21.1) | | | | |
| | Baseline: No subsidy | 4.4[10] (3.5–5.6) | 13.9[10] (11.0–17.4) | 2.2[7] (1.2–3.2) | | | | |
| **Tanzania—Mainland** <br><br> **Outlet survey (sales volumes); household survey (use)** | Intervention: CPM 2014 | | | 39.2*[7,9] | | | | |
| | Intervention: AMFm | 33.7~[10] | 61.5~[10] | 32.1*[7] (24–40.3) | | | | |
| | Baseline: No subsidy | 37.9[10] | 63.1[10] | 2.2[7] (1.1–3.3) | | | | |
| **Tanzania—Zanzibar** <br><br> **Outlet survey** | Intervention: AMFm | | | 60.7*[7,9] | | | | |
| | Baseline: No subsidy | | | 2.0[7] | | | | |
| **Uganda** <br><br> **Outlet survey** | Intervention: CPM 2015 | | | 47.5[7,8,9] | | | | |
| | Intervention: CPM 2013 | | | 43.0*[7,8,9] | | | | |
| | Intervention: AMFm | 44.2*[10] (37.7–50.8)[0] | 82.6~[10] (79.1–85.6) | 38.5*[7] (31.5–45.5) | | | | |
| | Baseline: No subsidy | 20.2[10] (15.1–26.5) | 40.1[10] (31.0–49.9) | 5.1[7] (2.5–7.7) | | | | |
| **Fink 2013 [34] Uganda** <br><br> **Household survey** | Intervention: AMFm | 47.5*[10] (45.4–49.5) | 65.3* (63.9–66.8) | | | | | |
| | Baseline: No subsidies | 37.2[10] (34.7–39.7) | 50.8 (48.9–52.8) | | | | | |

*(Continued)*

**Table 2.** (Continued)

| First author, Yr published; Country; Data source | Study Arms | ACT uptake | | | Antibiotic uptake | Patient adherence to treatment regimen | Medicine quality | Health outcomes |
|---|---|---|---|---|---|---|---|---|
| | | % febrile patients receiving ACT | % febrile patients receiving an antimalarial that received an ACT | % of antimalarial sales volumes that were ACTs | % of febrile patients that received an antibiotic | % of patients dispensed ACT who complete the full dose as directed | % of antimalarials purchased or sold that meet quality standards | |
| **Fiore 2018** [35] **Pooled** <br><br> **Household survey** | Intervention: AMFm/CPM | 9*[20] | | | | | | |
| | Control: No subsidies | 4 [20] | | | | | | |
| **Thomson 2014** [36] **Tanzania** <br><br> **Outlet survey (sales volumes); household survey (use)** | Intervention: AMFm | 26.9*[11] | 52.7*[11] | 34.0*[12] | | | | |
| | Baseline: No subsidy | 18.5[11] | 30.6[11] | 2.2[12] | | | | |
| **1.3 Interventions to enhance user adherence to subsidised ACT** | | | | | | | | |
| **Bruxvoort 2014** [37] **Tanzania** <br><br> **Client follow-up survey** | Intervention: Text message reminders to PMR staff | | | | | 68.3 (23.4) | | |
| | Control: no text message reminders | | | | | 69.8 (20.9) | | |
| **Cohen 2018** [38] **Uganda** <br><br> **Client follow-up survey** | Intervention: Social marketing packaging | | | | | 61.1 [2.6][13] | | |
| | Intervention: stickers on manufacturer's package | | | | | 69.5 [3.0][13, 14] | | |
| | Control: Manufacturer's package | | | | | 63.8 | | |
| **Raifman 2014** [39] **Ghana** <br><br> **Client follow-up survey** | Intervention: Long text message reminder to user | | | | | 62.3[15] | | |
| | Intervention: Any text message reminder to user | | | | | 57.1[15] | | |
| | Control: No text message | | | | | 58.0[15] | | |
| **4. Broader private sector strategies including ACT** | | | | | | | | |
| **Bjorkman 2021** [40] **Uganda** <br><br> **Household survey (use); mystery shoppers (medicine quality)** | Intervention: Villages with CHWs as retailers | 32.6[10, 13] | | | | | 90.6*[13] | |
| | Control: Villages without | 35.0[10] | | | | | 73.7[13, 16] | |

*(Continued)*

**Table 2.** (Continued)

| First author, Yr published; Country; Data source | Study Arms | ACT uptake | | | Antibiotic uptake | Patient adherence to treatment regimen | Medicine quality | Health outcomes |
|---|---|---|---|---|---|---|---|---|
| | | % febrile patients receiving ACT | % febrile patients receiving an antimalarial that received an ACT | % of antimalarial sales volumes that were ACTs | % of febrile patients that received an antibiotic | % of patients dispensed ACT who complete the full dose as directed | % of antimalarials purchased or sold that meet quality standards | |
| **Bjorkman 2019** [41] **Uganda**<br><br>**Household survey** | Intervention: Villages with CHWs as retailers | 67.2~[10, 13] | | | | | | 13.45~[17] |
| | Control: Villages without | 66.[10] | | | | | | 19.4 |
| **Thomson 2018** [42] **Tanzania**<br><br>**Outlet survey** | Intervention: Areas with drug shop accreditation | | | 41.6[18] | | | | |
| | Control: Areas without drug shop accreditation | | | 25.1[18] | | | | |
| **Briggs 2014** [43] **Tanzania**<br><br>**Exit interviews** | Intervention: Areas with drug shop accreditation | 24.6~[13, 19] | | | 11.0~[13, 19] | | | |
| | Control: Areas without drug shop accreditation | 16.6[13, 19] | | | 13.4[13, 19] | | | |

\* p<0.05 for significance of difference from control or baseline shown in table;

\*^ p<0.05 for significance of difference from baseline data which is not shown;

~ significance not reported

[1] No studies were identified reporting on outcomes related to referral on the basis of protocol

[2] Outcome is measured among febrile children 3–59 months using AL.

[3] % of mystery shoppers dispensed AL.

[4] % of volumes sold that were subsidised AL. Comparison is from period without subsidised ACTs (2012) to a period with subsidised ACTs (2013).

[5] Outcome is not presented for all age groups combined.

[6] Measured among drug shops.

[7] Outcome is % of volumes sold that were quality-assured ACTs among all private for-profit outlets, including facilities.

[8] Approximate reading from figure

[9] Statistical test is for change is from previous survey, not baseline.

[10] Outcome is measured among children <5

[11] Outcome is measured among febrile patients that sought care at a specialised drug seller.

[12] Outcome is % of volumes sold that were quality-assured ACTs among specialised drug sellers.

[13] Author's calculation

[14] Result are pooled for two versions of the sticker message. The sticker message "Malaria is not gone until. . ." had a larger effect on adherence.

[15] Results shown for drug shop clients only. For all provider types adherence was 61.5% in the control group, 66.4% for the short text message and 64.1% for the long text message.

[16] % of incumbent drug shops selling ACT that passes quality testing.

[17] Outcome is under 5 mortality per 1,000 years of exposure.

[18] % of sales volumes in drug shops that were AL.

[19] Outcome is measured among drug shop clients

[20] Outcome is percent of children 0–59 months with fever that took private sector ACTs

**Table 3. Study outcomes—Introducing and enhancing RDT and ACT use, and introducing and enhancing iCCM—% (95% CI where available).**

| Study Design | | RDT uptake | ACT uptake | Antimalarial dispensing according to RDT result | | | Antibiotic uptake | | | Patient adherence |
|---|---|---|---|---|---|---|---|---|---|---|
| First author, Yr published; Country; Data source | Study Arms | % febrile patients at study outlet receiving RDT | % febrile patients receiving ACT | % patients with negative test not receiving antimalarial | % patients with positive test receiving ACT | % patients treated according to test result | % patients with negative RDT receiving antibiotic | % patients with positive RDT receiving antibiotic | % all febrile patients receiving antibiotic | % patients dispensed ACT who complete full dose |
| **2. Introducing and enhancing RDT and ACT use** | | | | | | | | | | |
| **2.1 RDTs conducted by PMRs** | | | | | | | | | | |
| **Ansah 2015 [44] Ghana** | Intervention: Trained providers and free RDTs | 100 (provided to all) | 47.2[2~] | 97 | 92[1] | not stated | 0.6 | 0 | not stated | 95 |
| Provider records; Follow-up survey (adherence only) | Control: Trained providers but no RDTs | (RDTs not available) | 83.2[2] | | | | | | | No sig. diff from intervention (% not stated) |
| **Cohen 2015 [45] Uganda** | Intervention: Trained providers & subsidized RDTs | 17.5[1] | 29.3[2] | 56.3 | 40.9 | not stated | 35[1] | 31[1] | not stated | |
| Household survey; Study staff records (dispensing only) | Control: No intervention | 9.9 | not stated | not stated | not stated | not stated | not stated | not stated | not stated | |
| **Maloney 2017 [46] Tanzania** | Intervention: Trained providers and subsidized RDTs | 67*^ (58–71) | 32.4[2] | 90.8[2] | 84.4[2] | not stated | 10.5[2] | 15.6[2] | not stated | |
| Exit interviews | Intervention: Trained providers and unsubsidized RDTs | 66*^ (59–72) | 32.2[2] | 95.0[2] | 67.3[2] | not stated | 10.0[2] | 5.7[2] | not stated | |
| | Control: No training or RDTs | 3 | 43.2[2] | | | | | | | |
| **Mbonye 2015[8] [47] Uganda** | Intervention: Trained providers and subsidized RDTs | 97.7 | 60.8 | 98.5[4] | 99.0[4] | 98.8[4] | 46.0 (Hopkins 2017) | 23.6 (Hopkins 2017) | 34.9* (Hopkins 2017) | |
| Provider records; Follow-up survey (antibiotics only) | Control: Trained providers but no RDTs | | 99.7 | | | | | | 19.4 (Hopkins 2017) | |

(*Continued*)

**Table 3.** (Continued)

| First author, Yr published; Country; Data source | Study Arms | RDT uptake | ACT uptake | Antimalarial dispensing according to RDT result | | | Antibiotic uptake | | | Patient adherence |
|---|---|---|---|---|---|---|---|---|---|---|
| | | % febrile patients at study outlet receiving RDT | % febrile patients receiving ACT | % patients with negative test not receiving antimalarial | % patients with positive test receiving ACT | % patients treated according to test result | % patients with negative RDT receiving antibiotic | % patients with positive RDT receiving antibiotic | % all febrile patients receiving antibiotic | % patients dispensed ACT who complete full dose |
| **Dieci, 2023** [48] **Kenya** **Digital sales data** | Intervention: Patient subsidies—90% RDT subsidy, 80% ACT subsidy | 35.3*[7] | 72.0*[7] | 90.2* | 90.8* | | | | | |
| | Intervention: Provider incentives—USD 0.86 for RDT, USD 0.76 for ACT with positive RDT, USD 0.29 for recording | 27.7*[7] | 77.6[7] | 89.3* | 85.7* | | | | | |
| | Combined: Patient 60% RDT subsidy, 60% ACT subsidy; Provider incentive USD 0.14 for RDT, USD 0.14 for ACT with positive RDT, USD 0.29 for recording | 27.9*[7] | 73.1*[7] | 82.2* | 87.4* | | | | | |
| | Control: No intervention | 8.1[7] | 86.7[7] | 24.2 | 67.9 | | | | | |
| **Omale 2021** [49] **Nigeria** **Household survey** | Intervention: Social group meetings + provider training | 64.1*[3] | | | | | | | | |
| | Intervention: Social group meetings | 62.9*[3] | | | | | | | | |
| | Control: No intervention | 37.9[3] | | | | | | | | |
| **Onwujekwe 2015** [50] **Nigeria** **Results presented for PMRs only exc. public facilities**[3] **Exit interviews; Provider records (test results)** | Intervention: Trained providers, subsidized RDTs, plus school intervention | 8.4~[2] | 55.9~[2] | 13~[2] | 71~[2] | not stated | 17~[2] | 11~[2] | not stated | |
| | Intervention: Trained providers and subsidized RDTs | 12.3~[2] | 48.5~[2] | 43~[2] | 80~[2] | not stated | 17~[2] | 15~[2] | not stated | |
| | Intervention: RDT demonstration and subsidized RDTs | 24.8[2] | 47.5[2] | 45[2] | 76[2] | not stated | 12[2] | 17[2] | not stated | |
| **Soniran 2022** [51] **Ghana** **Mystery client survey** | Intervention: Trained providers and subsidized RDT | 38.1 (NB low n = 42) | 33.3 (NB low n = 42) | 83.3 (NB low n = 12) | not shown as n<10 | not shown as n<10 | | | | |
| | Control: No intervention | 23.3 (NB low n = 30) | 53.3 (NB low n = 30) | not shown as n<10 | not shown as n<10 | not shown as n<10 | | | | |

*(Continued)*

**Table 3.** (Continued)

| Study Design | | RDT uptake | ACT uptake | Antimalarial dispensing according to RDT result | | | Antibiotic uptake | | | Patient adherence |
|---|---|---|---|---|---|---|---|---|---|---|
| First author, Yr published; Country; Data source | Study Arms | % febrile patients at study outlet receiving RDT | % febrile patients receiving ACT | % patients with negative test not receiving antimalarial | % patients with positive test receiving ACT | % patients treated according to test result | % patients with negative RDT receiving antibiotic | % patients with positive RDT receiving antibiotic | % all febrile patients receiving antibiotic | % patients dispensed ACT who complete full dose |
| **2.2 RDTs conducted by study staff** | | | | | | | | | | |
| **Cohen 2015**[14] [52] **Kenya** <br><br> **Study staff records (RDT and antimalarials dispensed); Household survey (any ACT, any antibiotic)** | Intervention: Any ACT subsidy | 20.0 | 41.5 (92% sub)* ~ 35.1 (88% sub)* ~ 36.8 (80% sub)*[4] | 30 (92% sub) ~ 40 (88% sub) ~ 45 (80% sub)~[1] | 98 (92% sub)~ 98 (88% sub)~ 98 (80% sub)~[1] | not stated | | | 13.9 (92% sub) 12.3 (88% sub) 8.5 (80% sub)*[4] | |
| | Intervention: Any RDT subsidy | 29.1* | 40.7[4] | not stated | not stated | not stated | | | - | |
| | Control: No ACT subsidy | 21.4 | 19.0[4] | | | | | | 18.5[4] | |
| | Control: No RDT subsidy | 7.6 | 38.9[4] | | | | | | - | |
| **Ikwuobe 2013** [53] **Nigeria** <br><br> **Study staff records** | Intervention: Free RDTs | 100 (provided to all) | 42.0~[7] | 48.4 | 85.7 | not stated | | | | |
| | Control: No intervention | (RDTs not available) | 70.7[7] | | | | | | | |
| **Laktabai 2020** [54] **Kenya** <br><br> **Study staff records (RDT); exit interviews (medicines dispensed)** | Intervention: 50% RDT subsidy and 100% ACT subsidy | 98.6*[9] | 15.7~ | 94.9[4] | 77.4 | 92.2[4] | | | | |
| | Intervention: 50% RDT subsidy and 67% ACT subsidy | 100*[9] | 23.7~ | 93.9[4] | 87.0 | 92.4[4] | | | | |
| | Intervention: No RDT subsidy and 100% ACT subsidy | 96.2 | 26.8~ | 93.5[4] | 92.2 | 93.2[4] | | | | |
| | Intervention: No RDT subsidy and 67% ACT subsidy | 96.5 | 27.2 | 91.0[4] | 84.8 | 89.5[4] | | | | |
| **Modrek 2014** [55] **Nigeria** <br><br> **Follow-up survey** | Intervention: Text messages to adult patients, free RDT and ACT | 100 (provided to all) | | 76.9*[5] | | 79.8*[5] | | | | |
| | Control: Free RDT and ACT | 100 (provided to all) | | 58.5[5] | | 65.5[5] | | | | |
| **Saran 2016** [56] **Uganda** <br><br> **Patient follow-up survey** | Intervention: Free RDT and ACT subsidy | | | | | | | | | No sig. diff from control (% not stated) |
| | Intervention: ACT subsidy | | | | | | | | | 66.5 |
| **2.3 RDTs conducted by CHWs, with medicines provided by PMRs** | | | | | | | | | | |

(Continued)

**Table 3.** (Continued)

| First author, Yr published; Country; Data source | Study Arms | RDT uptake | ACT uptake | Antimalarial dispensing according to RDT result | | | Antibiotic uptake | | | Patient adherence |
|---|---|---|---|---|---|---|---|---|---|---|
| | | % febrile patients at study outlet receiving RDT | % febrile patients receiving ACT | % patients with negative test not receiving antimalarial | % patients with positive test receiving ACT | % patients treated according to test result | % patients with negative RDT receiving antibiotic | % patients with positive RDT receiving antibiotic | % all febrile patients receiving antibiotic | % patients dispensed ACT who complete full dose |
| O'Meara 2016 [57] Kenya — CHW records (RDT); follow-up survey (medicines dispensed) | Intervention: Free RDT and 50% ACT subsidy | 71.1~ | 43.9~ | 72.5[4] | 81.8*[10] | 76.2[4] | | | | |
| | Intervention: Free RDT and no ACT subsidy | 67.0~ | 33.0~ | 80.0[4] | 71.4 | 77.6[4] | | | | |
| | Intervention: No RDT subsidy and 50% ACT subsidy | 42.5~ | 32.7~ | 87.1*[4, 9] | 84.0*[10] | 85.7*[4] | | | | |
| | Intervention: No RDT subsidy and no ACT subsidy | 42.0 | 29.0 | 92.6*[4, 9] | 58.3 | 76.5[4] | | | | |
| O'Meara 2018 [58] Kenya — Household survey[15] | Intervention: Free RDT and ACT subsidy | 55.0*[3] | | 70.1[4] | 90.0 | 88.5*[4] | 47.3 | 25.2 | 22.6 | |
| | Control: No intervention | 44.7[3] | | 54.4[4] | 83.8 | 80.7[4] | 69.3 | 30.2 | 26.0 | |
| **3. Introducing and enhancing iCCM** | | | | | | | | | | |
| Awor 2014 [59] Uganda — Exit interviews (RDT uptake, medicines dispensed); Observation (dispensing by test result) | Intervention: iCCM | 87.7* (79.0–96.4) | 80.7* | 91.0 (NB low n = 11) | 100 (NB low n = 33) | 97.6 | not stated | not stated | 60*[11] | |
| | Control: No intervention | 0 | 41.1 | | | | not stated | not stated | 73.5[11] | |
| Bagonza 2021 [60] Uganda — Provider records | Intervention: iCCM with peer supervision | | | | | 53.5–66.3[12] | | | | |
| | Control: Standard iCCM | | | | | 47.1–67.1[12] | | | | |
| Kitutu 2017 [61] Uganda — Exit interviews | Intervention: iCCM | 47.8* | 22.4 | not stated | not stated | not stated[13] | not stated | not stated | not stated | |
| | Control: No intervention | 0.39 | 14.3 | | | | not stated | not stated | not stated | |

(Continued)

**Table 3.** (Continued)

| Study Design | | Outcomes | | | | | | | | |
|---|---|---|---|---|---|---|---|---|---|---|
| First author, Yr published; Country; Data source | Study Arms | RDT uptake | ACT uptake | Antimalarial dispensing according to RDT result | | | Antibiotic uptake | | | Patient adherence |
| | | % febrile patients at study outlet receiving RDT | % febrile patients receiving ACT | % patients with negative test not receiving antimalarial | % patients with positive test receiving ACT | % patients treated according to test result | % patients with negative RDT receiving antibiotic | % patients with positive RDT receiving antibiotic | % all febrile patients receiving antibiotic | % patients dispensed ACT who complete full dose |
| **Mbonye 2020** [62] **Uganda** **Provider records** | Intervention: iCCM | 86.9* | 33.1*[6] | 87.4 | 94.3*[6] | not stated | not stated | not stated | not stated | |
| | Control: No intervention | 58.5 | 38.7[6] | 95.6 | 83.0[6] | not stated | not stated | not stated | not stated | |

* p<0.05 for significance of difference from control or baseline shown in table;

*^ p<0.05 for significance of difference from baseline data which is not shown;

~ significance not reported

No studies were identified reporting on outcomes related to medicine quality or health outcomes. Only one study (Ansah et al.) reported Referral according to protocol

[1] approximate reading from figure

[2] Not stated in original papers; taken from Visser et al. review, based on Visser et al.'s contact with original study authors

[3] For Omale 2021 the indicator reported is "% febrile patients receiving RDT from any provider"; for O'Meara 2018 the indicator reported is "% febrile patients receiving any malaria test from any provider"

[4] For Laktabai 2020, O'Meara 2016 and O'Meara 2018 the indicator reported is "% patients with negative RDT not receiving ACT"; For Cohen 2015 (Kenya) and O'Meara 2018 results are reported from any provider. For Mbonye 2015 the indicator reported is "% patients with negative RDT not receiving AL or rectal artesunate" and "% patients with positive test receiving AL or rectal artesunate"

[5] For Modrek 2014 the indicators reported are based on medicines taken by patients (not on those dispensed); the indicator "% patients with negative test not receiving antimalarial" is reported only for the 57% of patients who bought both an antimalarial and drugs to treat their symptoms

[6] For Mbonye 2020 the indicators reported are "% all patients receiving AL" and "% patients with positive test receiving AL"

[7] For Ikwuobe 2013 the indicator reported is "% patients seeking to purchase antimalarials receiving ACT" rather than of all febrile patients. For Dieci 2023 the denominator for these indicators is "all people who bought an antimalarial or an RDT"

[8] Also related studies by Hutchinson et al. 2017 (including follow up survey findings), Hansen et al. 2017

[9] Significance of difference between the subsidised RDT groups and the unsubsidised RDT groups, averaged across the price levels of ACTs

[10] Significance of difference between the subsidised ACT groups and the unsubsidised ACT groups, irrespective of whether they received an RDT subsidy

[11] For Awor 2014 the indicator is reported out of all sick children, not just those with fever (though 98% of children reported fever)

[12] Bagonza 2021 used interrupted time series–table includes range of results across 7 monthly post-intervention data points (no statistical difference in trend between intervention and control)

[13] Kitutu 2017 do not report this indicator but do report an alternative indicator of appropriate treatment for malaria (see text)

[14] Cohen 2015 Kenya results for the 11 treatment arms are not presented separately. Rather the effect of ACT and RDT subsidies is estimated by pooling arms in different combinations

[15] Antibiotic uptake results for this study are available in both O'Meara 2018 and Laktabai 2022; we present the results from O'Meara 2018 as they relate to the final point of evaluation. The direction and size of effects reported are similar in the two papers.

combined with branding of the subsidised product, behaviour change communications, PMR training (one day where duration stated), and recommended retail prices (RRP). Kangwana et al. (Kenya) and Lussiana et al. (Angola) involved subsidies of paediatric ACT only [28, 30], while Sabot et al. (Tanzania) and Talisuna et al. (Uganda) subsidised ACTs for all age groups [31, 32]. In 3 studies, the subsidised ACTs were delivered directly to participating PMR, while in Sabot et al. they were provided through a designated wholesaler. The studies covered a range of low, moderate and high malaria transmission settings, with three in rural areas, and

one covering urban and rural areas. The number of participating PMRs ranged from 151 to 225 (not reported in one study). In all studies eligible PMR included drug shops and/or pharmacies, while Kangwana et al. also included general shops. Three studies were pre-post evaluations: of these, two included a control group [31, 32], and one did not [30]. The fourth study was a cluster randomized control trial [28].

There were large increases in ACT uptake across all studies with, for example, a 26.4 percentage point difference in febrile patients receiving ACT between the intervention and control arms in Kenya (household survey) [28], and a 20.6 percentage point difference in Uganda (exit interviews) [32]. In Tanzania the market share of ACT among PMR antimalarial sales was negligible in control areas and over 60% at endline (outlet survey) [31]. Where measured, these effects were observed across all socio-economic groups, with ACT uptake higher in intervention areas among all quintiles in Kenya [28], and even improving more among households with less education in Uganda [32]. Changes in other outcomes relevant to this review were generally not measured, with the exception of patient adherence to ACT dosing in Kenya, where the difference was not statistically significant (household survey) [28].

It was not possible to attribute variations in intervention effectiveness across studies to specific intervention components, due to the small number of studies, similarity of programmes, and differences in data collection methods (S6a Table). The Tanzanian study implemented an RRP in one of the two intervention areas with mixed results: uptake was higher in the RRP area for children but not for adults, reflecting the finding that for adult doses the RRP appeared to have had the unintended effect of artificially inflating prices above the market rate [31].

**National ACT subsidy programmes.** Five studies explored the impact of national ACT subsidy programmes–the AMFm and its successor the Co-Payment Mechanism (CPM). The AMFm covered 7 African countries: Ghana, Kenya, Madagascar, Niger, Nigeria, Uganda, and Tanzania (mainland and Zanzibar) from 2010–2012 [12]. It involved price negotiations with manufacturers, and subsidy of quality-assured ACTs at the "factory-gate". Subsidised ACTs were imported by registered first-line buyers, and then distributed to PMRs through their usual distribution chains. Supporting interventions at country level involved some combination of behaviour change communications, provider training, and RRPs, with considerable variation in their duration and scale across countries. Following a transition period in 2013, the CPM replaced AMFm in 6 countries (Ghana, Kenya, Madagascar, Nigeria, Tanzania, Uganda), involving similar intervention components to AMFm, but with differences in the source of funds, process of approving orders, and lower subsidy levels [8].

Three studies assessed the impact of AMFm only, including the multi-country AMFm Independent Evaluation [12, 33], and two single country studies by Fink et al. in Uganda and Thomson et al. in Tanzania [34, 36]. Two later multi-country studies covered both the AMFm and CPM periods [8, 35]. The studies covered a range of low, moderate and high malaria transmission settings. Four of the studies were pre-post comparisons without control, while Fiore et al. conducted a pre-post evaluation comparing countries with and without AMFm/CPM [35].

The studies found increases in the market share of quality-assured ACTs in private for-profit outlets across all countries implementing these subsidy programmes (outlet surveys). In the five countries with data from the CPM period (Kenya, Madagascar, Nigeria, Tanzania, Uganda) increased market share under AMFm was either sustained or rose further, except in Madagascar, where it increased substantially in the initial CPM period, but then fell to pre-AMFm levels [8]. The magnitudes of market share increases from pre-AMFm to the most recent data point were over 30 percentage points everywhere but Madagascar and Niger.

Data on the impact on ACT use from household surveys are somewhat more limited, and may also reflect concurrent changes in public sector ACT provision. Pre-post data from Madagascar, Nigeria and Uganda show increases in ACT use among febrile children under 5,

reflecting substantial substitution away from other antimalarials [33]. In Tanzania, national data show a decrease in the percent of febrile patients receiving ACT overall following AMFm introduction [33], but more detailed sub-national data indicate that this partially reflected reduced use of ACTs in public facilities associated with an increase in diagnostic testing, while ACT use among patients visiting PMR increased [36]. Fiore et al. compared changes over time in ACT use among febrile patients that sought care in the private sector in AMFm/CPM countries (Ghana, Nigeria, Uganda) and 12 comparator countries [35]. They found that AMFm/CPM increased use overall, but this seems to be mostly driven by improvements in Uganda. No other outcomes relevant to this review were reported by these studies.

As with the sub-national pilots, there was no evidence that increases in ACT uptake were limited to certain socio-economic groups. Increases in market share and use were similar in urban and rural areas [12, 35]; and where overall ACT use was shown to have increased (Madagascar, Nigeria, Uganda), this was also documented among children in the poorest two quintiles [33, 34]. However, in Nigeria the largest increases in market share were recorded among the top two quintiles [33].

It is challenging to identify the contribution of specific intervention components to performance (S6a Table). For example, Kenya and Tanzania had sustained implementation of communications activities and adherence to RRPs seemed strong, and both countries saw large, sustained market share increases. However, large increases were also seen in Uganda, despite poor adherence to RRPs, and very limited behaviour change communications. The Independent Evaluation attributed smaller market share increases in Madagascar and Niger during AMFm to low ordering, political and economic disturbances, and the high proportion of general retailers among PMRs who may have been viewed as less appropriate ACT sellers than pharmacies/drug shops [33]. However, much higher market share was achieved in Madagascar under the CPM in 2013 [8], indicating that their retail market composition was not necessarily a barrier to higher ACT uptake.

**Interventions to enhance user adherence to subsidised ACT.** Three RCTs evaluated interventions to improve adherence to the ACT dosing regimen (% patients dispensed ACT who complete the full dose as directed). In Tanzania, Bruxvoort et al. sent regular text messages to PMR staff to remind them about the correct dispensing advice to give patients [37]. In Ghana, Raifman et al. sent text messages to patients for 3 days after their visit to a facility or PMR to remind them to complete the full course, with patients randomly allocated to receive either a long or short message [39]. In Uganda, Cohen et al. modified the ACT packaging to provide further information on adherence either through additional social marketing packaging or a simpler sticker attached to existing packaging [38]. These were all short, focused interventions (2.5–5 months) involving 9–84 providers, in moderate to high transmission settings, and in a mix of urban and rural, or rural only areas. Adherence was measured at follow-up visits to patient in their homes.

Improvements in adherence either did not occur or were modest (less than 6 percentage points difference between intervention and control) across the three studies. In Uganda, the simpler sticker was found to be more effective than more complex social marketing packaging [38]. In Ghana, results from drug shops only showed no significant effect for either text message, though results for all provider types combined (drugs shops and facilities) indicated that the shorter text message improved adherence while the longer message did not [39].

## Introducing and enhancing RDT and ACT use

15 studies were identified on introducing or enhancing RDT use in PMRs, all but one [51] of which either also included ACT subsidies or were conducted in the context of a national ACT

subsidy programme. We present the studies in two groups: the first group covers 13 studies where RDTs were conducted at the PMR, while the second group covers 2 studies where community health workers (CHWs) conducted RDTs and provided vouchers for ACT purchase at PMRs in the same area [57, 58] (Table 3).

**RDTs conducted at PMRs.** Of the 13 studies where RDTs were conducted at PMRs, most involved drug shops with some including pharmacies. Reflecting very low routine RDT provision by PMRs in most settings, nine studies involved introduction of RDTs to PMRs where they were not previously used, while only 4 explored interventions to strengthen existing RDT use by PMRs [45, 48, 49, 51]. The intervention package also involved ACT subsidies in 6 studies [47, 48, 52, 54–56], while in 5 of the remaining 7, AMFm or CPM subsidised ACTs were available in the market [44–46, 50, 53]. Eight studies tested an intervention package with PMR staff conducting the RDTs [44–51]; while in 5 studies the RDTs were conducted by research team members stationed at the PMR [52–56]. Of the latter group 3 studies generally had relatively short durations of 4 months or less (S5 Table).

For the 8 studies where PMR staff conducted the RDTs, the interventions involved PMR training (1–4 days) and supervision (weekly, monthly or quarterly). In addition, Dieci et al. tested the provision of performance incentives to pharmacy staff for providing RDTs, providing ACT conditional on a positive test, and appropriate record keeping [48]. Five studies included communications activities: Ansah et al., Mbonye et al. and Soniran et al. conducted community meetings [44, 47, 51]; Omale et al. tested sensitisation and education of women's and men's social groups [49]; and Onwujekwe et al. included a school intervention [50]. In 5 studies RDTs were provided directly to PMRs by the study team [44, 47, 48, 50, 51], in 2 studies PMRs purchased RDTs through designated wholesalers [45, 46], and in one study RDTs were obtained through regular distribution channels [49]. In one study RDTs were provided free to users [44]; in most others RDTs were subsidised, except for Omale et al. and one intervention arm in Maloney et al. and Dieci et al. where no subsidy was provided [46, 48, 49]. The number of providers included varied from 7 [44] to 262 [46], and the duration in months from 3 [49] to 17 [44]. All the studies were conducted in rural areas, while 3 also included urban areas, and all in situations of moderate or high malaria transmission, except Dieci et al. (mainly low transmission) [48]. All studies were cluster randomised trials, with the exception of Maloney et al. and Soniran et al., which were pre-post evaluations comparing intervention and control districts [46, 51].

The five studies where RDTs were conducted by research team members were not aiming to evaluate a comprehensive RDT intervention package, but rather to test specific hypotheses about user behaviour in a small number of PMRs (ranging from 1–10 retailers). Laktabai et al. and Cohen et al. studied the impact of varying the level of RDT and ACT subsidy, with for Laktabai et al. the ACT subsidy being conditional on a positive RDT result [52, 54]. The other 3 studies in this group provided RDTs for free: Saran et al. investigated whether having an RDT increased patient adherence to the full treatment dose; Modrek et al. investigated the effect of text message reminders on whether patients later consumed the correct medicine for their RDT result; and Ikwuobe et al. simply assessed medicines dispensed when an RDT was conducted [53, 55, 56]. Three of these five studies were in rural and two in urban areas, and all under moderate or high malaria transmission. Four studies individually randomised the intervention at the patient level, while Ikwuobe et al. compared one intervention and one control PMR [53].

*RDT uptake.* RDT uptake refers to the percent of febrile patients at the outlet receiving an RDT. In 11 of the studies RDT uptake was zero or minimal in the absence of any intervention, but in 4 studies RDTs were already used, ranging from 8% of pharmacy clients in Kenya (digital sales records) [48], to 10% of drug shops in Uganda (household survey) [45], 23% of drug

shops in Ghana (mystery shoppers) [51], and 38% of all providers in Nigeria (household survey—data not provided by authors for PMR clients only) [49].

In two of the studies where PMRs conducted RDTs by Ansah et al. and Mbonye et al., RDT uptake following the intervention was reported to be at or close to 100%, though both relied on the PMRs' own records, raising the risk that staff will be more likely to record patients who were tested, or of other Hawthorne effects [44, 47]. For the 6 other studies where PMRs conducted RDTs, uptake in the intervention group was close to two thirds for Maloney et al. (exit interviews) [46] and Omale et al. (household survey) [49], close to one third for Dieci et al. (digital sales data) [48] and Soniran et al. (mystery shoppers) [51], but under 20% for Onwujekwe et al. (exit interviews) [50] and Cohen et al. Uganda (household survey) [45]. Two of the studies where research staff conducted RDTs reported on uptake based on study team records, with results over 96% in Laktabai et al., but under 30% in Cohen et al. [52, 54] (in Modrek et al., Saran et al. and Ikwuobe et al. study staff provided RDTs to all intervention patients so uptake is not presented).

*Dispensing according to test result*. Our standard definition of correct dispensing by RDT result was a patient with a negative test receiving no antimalarial, and one with a positive test receiving an ACT, although there was variation in these indicators across studies. For example, some studies defined correct dispensing for negatives as not receiving ACT rather than not receiving any antimalarial. Of the 9 studies reporting rates of correct dispensing for positive RDTs, Ansah et al. and Mbonye et al. had very high rates of 92% and 99% respectively based on provider records [44, 47]. Rates over 80% were also seen in at least one arm in 3 studies based on exit interviews [46, 50, 54], two studies based on study staff records [52, 53], and one based on digital sales data [48], but Cohen et al. reported a much lower rate of 41% based on household survey data [45].

Of the 11 studies reporting rates of correct dispensing for negative RDTs, 5 reported rates over 90% in at least one arm including the two studies based on provider records [44, 47], two based on exit interviews [46, 54], one based on study staff records [52], and one based on digital sales data [48]. However, the other studies encompassed a wide range from 83% (mystery shoppers–though sample size very low) [51], to 77% (patient follow up survey) [55], 56% (household survey) [45], 48% (study staff records) [53] and 13–43% (exit interviews) [50]. Whether RDTs were performed by PMR or study staff did not appear to have a systematic effect on rates of correct dispensing for positive or negative RDTs (S6b Table).

Ansah et al. and Mbonye et al. also assessed dispensing in relation to microscopy results based on samples collected at the same time as the RDT test but examined at a later date in research team laboratories [44, 47]. As these data can be presented for all patients regardless of whether they received an RDT, they can be used to assess targeting of medicines in the intervention and control/baseline in relation to true parasitological status. Both studies reported substantially higher rates in the intervention than control arm of correct dispensing by microscopy results (slide positive receives ACT and slide negative does not receive antimalarial), with significant differences of 47 percentage points [44] and 39 percentage points [47]. In the intervention arms rates of correct dispensing by microscopy result were lower than when based on RDT result, despite extremely high RDT uptake in these two studies. This reflected low specificity of RDTs compared to microscopy of 63% [47] and 70% [44] i.e. RDTs led to a high number of false positives. While this could partially reflect the detection of HRP2 antigen persistence by RDTs, Ansah et al. note that specificity was particularly low in three drug shops, raising the possibility that some PMRs were incorrectly reporting positive RDT results to justify antimalarial sales [44].

*ACT uptake*. A comparison of ACT uptake (% all febrile patients receiving ACT) with and without RDT introduction was reported in 5 studies. While ACT uptake is not an indicator of

quality of care as the latter depends on whether the patients are parasitaemic, ACT uptake is a useful indicator of potential savings in ACT from RDT introduction. Five studies showed substantially lower ACT uptake in the intervention arm compare to the control: Ansah et al. of 83% v. 47% (provider records), Mbonye et al. of 99.7 v. 61% (provider records), Ikwuobe et al. of 71 v. 42% (study team records), Dieci et al. of 87% v. 72–78% (digital sales data), and Soniran of 53% v. 33% (mystery shoppers) [44, 47, 48, 51, 53]. It was notable that Ansah et al., Mbonye et al. and Dieci et al. reported very high rates of ACT uptake in control arms without RDTs (>83%). Maloney et al. (exit interviews) and Cohen et al. (household survey) showed no significant difference in ACT uptake between arms, but the levels in their intervention arms (32% and 35–41% respectively) [46, 52] were lower than in the intervention arms of the five studies showing large reductions.

*Other outcomes*. Antibiotic uptake was reported in six studies post-intervention, for RDT negatives ranging from 0.6% (provider records) [44] to 46% (provider records) (Mbonye et al., reported in Hopkins 2017 [63]), and for RDT positives from zero (provider records) [44] to 31% (household survey) [45]. Only Mbonye et al. reported on the difference in antibiotic uptake with and without RDT introduction, which they found to be 15.5 percentage points higher in the intervention arm (Mbonye et al., reported in Hopkins 2017 [63]). Two studies assessed the impact of RDT provision on patient adherence (% patients dispensed ACT who completed the full dose) but neither found a significant impact (patient follow-up surveys) [44, 56]. Only one study reported the proportion of patients where a referral was required who were referred; by protocol Ansah et al. required that all RDT negatives were referred to health facilities, and they found that this occurred in 80% of cases, of which 62% later reported that they went to the referral facility (patient follow-up survey) [44]. Other studies reported overall referral levels but these are hard to interpret without knowing whether referral was required. No studies reported on other review outcomes, or on variations in effectiveness by socio-economic status or urban/ rural location.

*Impact of variation in RDT intervention design*. The impact of variation in intervention design can be assessed either within studies with multiple intervention arms, or by comparing across studies with different intervention packages (Table 3 and S6b Table). When comparing across studies, there is some indication that lower RDT retail prices were associated with higher RDT uptake. The three studies with very high RDT uptake provided RDTs free [44] or had relatively low RRPs or actual prices of USD 0.21 [54] and USD 0.23 [47], compared with USD 0.5 [45] and USD 0.69 [50] in two of the studies with low uptake (all prices converted to 2020 USD). By contrast, Cohen et al. reported low uptake at a price of only USD 0.23 [52], but given that this was the earliest study identified (2009) low uptake could partially have reflected patients' unfamiliarity with RDTs.

Three studies assessed the impact of varying the RDT price as part of their study design. Maloney et al. found no difference in RDT uptake (exit interviews) between a subsidised RRP of USD 0.36 and unsubsidised of USD 0.75 [46]. Cohen et al. found no substantive difference in RDT uptake when comparing free RDTs with those subsidised at USD 0.23, but a significant increase in RDT uptake from 7.6 to 29.1% (study staff records) when comparing any subsidy (free or USD 0.23) with those unsubsidised at USD 1.51 [52]. In Laktabai et al. a subsidised price of USD 0.21 compared with an unsubsidised price of USD 0.41 had limited impact as RDT uptake was over 96% in all arms (study staff records) [54]. These varying results likely reflect the considerable variation in the level of the unsubsidised price across studies.

Two of these studies also explored variation in the ACT subsidy level. Laktabai et al. found that varying whether ACT was provided free or subsidised (conditional on a positive RDT) had no impact on RDT uptake or antimalarial dispensing, reflecting very high compliance with guidelines in all study arms (study staff records) [54]. Cohen et al. found varying the ACT

subsidy from 80 to 92% had no impact on ACT uptake overall, although it substantially increased compared to the unsubsidised price of USD 7.48 [52]. Varying the ACT subsidy from 80 to 92% had no impact on dispensing according to a positive RDT (which was high for all groups), but the lower ACT subsidy increased correct dispensing for RDT negative patients (no antimalarial) (study staff records).

Of the 6 studies where PMRs conducted the RDT there was some indication that interventions with higher training and supervision intensity had better dispensing outcomes [44, 47] (S6b Table), but this could also have reflected outcome measurement based on provider records. Notably Maloney et al. also reported high rates of appropriate dispensing (exit interviews) but had shorter training and less frequent supervision [46]. There was no clear pattern in RDT uptake across studies based on whether RDTs were directly delivered to PMRs by the study team (S6b Table).

Four studies explored variations in other supportive measures across study arms, with two testing the impact of adding provider training [49, 50], and one each the addition of social group mobilisation [49], a school intervention [50], text message reminders to users [55], and provider performance incentives [48]. Omale et al. found that the arm receiving social group mobilisation had higher RDT uptake (household survey) [49], and Modrek et al. found that the arm where patients received text messages reminding those RDT positive to take ACT and those RDT negative not to take any antimalarials were significantly more likely to consume antimalarials in line with their RDT result (patient follow-up survey) [55]. However, Onwujekwe et al. found no evidence that adding provider training or a school intervention improved RDT uptake or dispensing (exit interviews) [50], and Omale et al. found no difference in RDT uptake from adding provider training to social group mobilisation (household survey) [49]. Dieci et al. found no significant differences between 3 intervention arms comparing (i) patient subsidies for RDTs and ACTs (condition on a positive test), (ii) provider performance incentives for appropriate RDT use, ACT provision and record keeping, and (iii) a combined arm with both patient subsidies and provider incentives (digital sales data) [48].

**RDTs conducted by CHWs, with medicines provided by PMRs.** Two studies by O'Meara et al., both in Western Kenya, used a different intervention model, where CHWs located close to PMRs were trained to conduct RDTs and provided vouchers for ACT purchase at PMRs for those patients with a positive RDT result [57, 58]. Both studies were conducted in rural areas, with low or low to moderate malaria transmission. The earlier study was a 12-month individually randomised trial with a factorial design involving 11 PMRs [57]. Individuals with a malaria-like illness who had not yet sought treatment were recruited from their homes and allocated to receive an unsubsidised or fully subsidised RDT by a CHW and one of two levels of ACT subsidy. The later study was a 20-month cluster randomised trial involving 42 PMRs, comparing community clusters with CHWs providing free RDTs and conditional ACT subsidies with clusters where CHWs did not provide RDTs [58]. In both studies ACTs were supplied to PMRs through the standard supply chain. No training or supervision on malaria case management was provided to PMRs beyond a brief orientation on the vouchers and ACT dosing, but the later study included community level communications activities [58].

O'Meara et al. 2016 found that patients offered a free RDT had higher RDT uptake at CHWs, and higher malaria test uptake overall (RDT or microscopy), with an adjusted difference of 18.6 percentage points (patient follow-up survey) [57]. Receiving the conditional ACT subsidy did not lead to higher test uptake, but did increase dispensing in line with a positive RDT result by 19.5 percentage points (patient follow-up survey) [57]. Appropriate ACT use following a test was highest when clients paid for their RDT and were given a conditional voucher for their ACT. However, as RDT uptake was low when they were unsubsidised, overall targeting of ACTs in all participants was highest in the groups that received a free RDT. This

analysis informed the design of the intervention for the cluster randomised trial, which included a free RDT and conditional ACT subsidy [58]. Household survey data showed that at the community level this intervention led to significantly better results in the intervention arm compared to the control arm for test uptake (55% v 45%), and correct dispensing for those with a negative RDT (70% v 55%), and with a positive RDT (90% v 84%) [58]. The provision of antibiotics was slightly lower in the intervention arm (47% v 69% for RDT negatives; 25% v 30% for RDT positives) (significance not reported).

## Introducing and enhancing iCCM

Four studies were identified on introducing and enhancing iCCM in PMRs, all from Uganda (Table 3). The iCCM interventions studied covered the treatment of children under 5 years for malaria (using RDT and ACT), pneumonia and diarrhoea. Three studies evaluated the introduction of iCCM [59, 61, 62], and a fourth the addition of peer supervision to an existing iCCM programme [60]. All four studies included drug shops in rural areas, with one also including private clinics and urban areas [62]. All studies were in areas of moderate or high malaria transmission, with the exception of Kitutu et al., where transmission was low to moderate [61].

The three studies of iCCM introduction included one cluster randomised trial [62] and 2 pre-post evaluations with control [59, 61], with the interventions involving PMR training (4–5 days), supervision (weekly to start, or monthly), and communications activities (eg radio, public meetings, shop branding, posters, leaflets, and/or training CHWs or Village Health Teams). Supplies, including ACTs and RDTs, respiratory timers, and medicines for pneumonia and diarrhoea were distributed directly to PMRs [59, 62] or through a designated wholesaler [61]. RDTs were provided free to users and ACTs were subsidised [59, 61]. The number of intervention providers ranged from 44 [59] to 61 [61], and the intervention duration from 9 [59] to 26 months [62].

The three studies introducing iCCM to PMRs all showed substantial increases in RDT uptake. Awor et al. and Kitutu et al. had zero or negligible uptake in the control arm compared to uptake in the intervention arm of 88% and 48% respectively (both exit interviews) [59, 61]. In Mbonye et al. uptake was 59% in the control arm compared to 87% in the intervention arm (provider records) [62]. Awor et al. and Mbonye et al. both reported very high rates of dispensing on the basis of RDT result of 100% and 94% respectively for positive RDTs, and 91% and 87% for negative RDTs, though these figures are based on direct observation [59] and provider records [62]. Kitutu et al. did not report these indicators but did find a substantial difference in a composite "appropriate treatment" indicator based on testing and dispensing by protocol for both febrile and afebrile children between the intervention (57%) and control (1%) areas (exit interviews) [61]. The three studies report mixed findings on ACT uptake: Awor et al. show substantially higher uptake in the intervention arm (exit interviews) [59], Mbonye et al. show significantly lower uptake (provider records) [62], and Kitutu et al. no difference (exit interviews) [61]. The effect on antibiotic use is reported by Awor et al. only, showing a significantly lower uptake by 13 percentage points (exit interviews) [59].

The fourth iCCM study by Bagonza et al. was an interrupted time series evaluation of the use of peer supervision to strengthen iCCM in 60 rural drug shops [60]. Selected PMR staff received a 3-day refresher training, and then made monthly visits to their peers for 8 months [60], while all products were provided by standard supply chains. The peer supervision had no impact on dispensing by test result (provider records) (the effect on other outcomes is not reported).

No iCCM studies reported other review outcomes or on variation in effectiveness by socio-economic status or urban / rural location.

## Broader private sector strategies including ACT

Three studies were identified evaluating broader PMR interventions that included ACT provision. As none included RDTs, the results are presented in Table 2 which shows studies involving ACT subsidies to facilitate comparison across similar outcomes. Two studies evaluated the Accredited Drug Dispensing Outlet (ADDO) programme in Tanzania [42, 43], which had been implemented in rural and urban areas of 14 of Tanzania's 21 region at the time of the studies, covering around 3,800 PMRs. The ADDO programme involved an initial PMR training of 35 days and regular supervision. ADDOs were permitted to stock 49 prescription-only medicines, including ACT, which was subsidised through AMFm at the time of the studies. Both studies were post-intervention assessments comparing ADDO with non-ADDO regions. Briggs et al. covered one ADDO and one non-ADDO region, with moderate and low to moderate malaria transmission [43], while Thomson et al. was nationally representative covering areas with low, low to moderate, and moderate transmission [42]. The studies found weak evidence that the ADDO programme improved ACT uptake: Thomson et al. found market share to be 42% in ADDO and 25% in non-ADDO regions based on outlet survey data though the difference was of borderline significance [42]. Briggs et al. reported the proportion of febrile patients obtaining ACT to be 25% in the ADDO region compared with 17% in the non-ADDO region (exit interviews) [43].

The third study was a randomised controlled trial evaluating the establishment of the Living Goods model where, following a 2-week training, a cadre of CHWs were established as retailers in over 800 rural Ugandan villages with high malaria transmission, selling a package of health-related commodities including ACTs but not RDTs [41]. There was no evidence that the intervention improved ACT uptake (household survey) [40, 41]. However, it was found to have a "spillover effect" of improving the quality of ACTs sold to mystery shoppers by incumbent drug shops in the villages where the CHW retailers were located, with 91% of ACTs meeting quality standards in intervention villages, compared with 74% in the control [40]. In addition, the programme was found to improve under 5-mortality (19.4/1000 in control villages and 13.5/1000 in intervention areas), although this is likely due to the cumulative effect of the broader package of products, given that ACT uptake did not increase (household survey) [41].

## Discussion

This systematic review synthesises research findings across a wide range of interventions relevant to malaria case management in PMRs in sub-Saharan Africa, providing a holistic view of the evidence base on policy options for retail intervention. We adopted broader study design eligibility criteria than many systematic reviews, including studies with randomised and non-randomised control groups, and those with a baseline but no contemporaneous control. This partly reflected a desire to be as inclusive as possible of available evidence, but also the fact that randomised or even controlled designs may not be feasible or appropriate when evaluating large scale operational implementation of interventions such as AMFm, where distribution through standard supply chains or use of mass media severely limits the potential to restrict interventions to certain areas. We excluded studies that had neither a historical nor a contemporaneous control, though we recognise that uncontrolled cross-sectional studies can provide additional insights, and we draw on some of these in this Discussion. Our review also excludes qualitative research, though we recognise that this is vital for understanding the reasons for

intervention (in)effectiveness, and the broader intended and unintended consequences. Geographical coverage of included studies was quite broad, covering urban and rural areas, and 8 countries, though the only evidence from Francophone West and Central Africa was one study covering AMFm in Niger. This may present a barrier in terms of generalisability to the many high malaria burden countries in these regions such as DRC, Mali, Burkina Faso and Cameroon.

We summarise here the key findings under each intervention category, and identify the outstanding evidence gaps, before turning to recommendations for future research. Beginning with ACT subsidy programmes and their supporting interventions, there is strong evidence that these increase the market share of quality-assured ACT, with the increase substantial in most settings. This has been demonstrated through both controlled small-scale studies, and the evaluation of AMFm and CPM at national scale in multiple countries. Later studies have indicated that market share gains from AMFm were generally sustained even with lower CPM subsidy levels in most settings. Small-scale studies also provide strong evidence of increased ACT use based on household survey data. For national subsidy programmes household survey data are less complete and not so clear cut, but during AMFm ACT use increases were documented in 3 countries overall and among PMR users in a further one. There is also positive evidence on the equity of ACT subsidy programmes, with benefits experienced in both rural and urban areas, and among poorer groups.

A meta-analysis by Morris et al. of the impact of ACT subsidies on private sector ACT use provided further support for these conclusions, estimating that each USD 1 decrease in ACT price was linked to a 24 percentage point increase in ACT use in the private sector, with no significant differences in this relationship when comparing the poorest vs richest groups, rural vs urban populations or children vs adults [23]. Other cross-sectional studies not eligible for this review indicate that the coverage of ACT subsidies can vary across groups though this depends on context. Ye et al. found that the market share of AMFm-subsidised ACTs in private for-profit outlets was higher in remote than in non-remote areas in Ghana, but lower in remote areas in Kenya [64]. Tougher et al. explored the impact by socio-economic group, finding that in Nigeria use of subsidized ACTs from for-profit outlets was concentrated among the rich, while in Uganda it was concentrated among the poor [65].

No interventions leading to substantial improvement in user adherence to ACT dosing regimens were identified, with adherence remaining at around two thirds in nearly all studies, indicating the range of factors beyond information that influence user behaviour [66]. RDT provision also did not improve adherence [44, 56].

Turning to the literature on RDT and iCCM provision, there are similarities between these two groups of studies, although the iCCM studies are far fewer in number and all from Uganda. For both interventions most studies are small-scale pilots or experiments, with strong internal validity reflecting their individually or cluster randomized designs. However, the interventions often include features not replicable under implementation at large scale, such as providing ACT or RDT for free to PMRs and in some cases to users, direct distribution of commodities to PMRs, or frequent supervision, and in several studies research team staff administering the RDTs (S5 Table). Despite these limitations, one can draw a number of conclusions from this body of evidence. First provision of both RDT and iCCM by PMRs is feasible. Secondly, in two studies under relatively controlled conditions, very high RDT uptake (over 97%) and dispensing by RDT result (over 92%) were found, though these studies relied on data collection methods likely to be subject to substantial Hawthorne effects, such as PMRs' own records or direct observation. In studies under more operational conditions and with methods less subject to Hawthorne bias, rates of RDT uptake of around two thirds were found in some studies, though in others uptake was below 20%. The potential for reductions in price

to increase RDT uptake was mixed: reducing prices from high unsubsidised levels (USD 1.5 or above) to below USD 0.4 did stimulate uptake, but demand was relatively insensitive to smaller changes in subsidised prices. High rates of dispensing by RDT result were demonstrated in most relatively operational settings, especially for RDT positive patients where in all but one study over 80% obtained ACT. For RDT negatives, many studies found appropriate dispensing rates over 77%, though in two studies this was under 50%. It was notable that all but one of the studies testing RDTs in PMRs were conducted in moderate to high malaria transmission settings (S3 Table), though one might expect RDT use to be particularly important in curtailing unnecessary ACT use in lower transmission settings. Similar uptake and dispensing results were obtained where CHWs conducted RDTs and provided vouchers for ACT purchase at PMRs [57, 58]. This hybrid approach would only be feasible in settings with a substantial functional and sustainable CHW network, and would require engagement and coordination across a wide range of stakeholders. No evidence was reported on the equity impact of RDT or iCCM introduction.

There was wide variation in rates of antibiotic dispensing to febrile patients in RDT and iCCM studies, from close to zero to around a third, despite nearly all antibiotics being prescription-only medicines in most settings [7]. Managing antibiotic use is hampered by the lack of clear guidance on how non-malaria febrile illnesses should be treated/referred by PMRs [10]. In particular, concern has been expressed that introduction of RDTs could lead to antimalarials being substituted by antibiotics for RDT negatives, with overall increases in antibiotic use documented in many public sector settings [63]. Within this review two studies showed little change or a decrease in antibiotic provision [58, 59], while one showed an increase (Mbonye et al., reported in Hopkins 2017 [63]), though it was not possible to assess for which patients antibiotics were warranted.

In marked contrast to the ACT subsidy literature, there were no eligible studies on RDT or iCCM implementation at large scale, with the biggest being Maloney et al., covering 2 districts in Tanzania [46]. This reflected a lack of eligible evaluations for some of the larger-scale RDT and iCCM initiatives in sub-Saharan Africa. A prominent example is the USD 20 million UNITAID-funded project to create private sector RDT markets in five African countries between 2013 and 2016 [67, 68]. In Kenya, Madagascar and Tanzania the implementing agency procured RDTs and provided a range of services along the supply chain. In Nigeria and Uganda, both RDTs and ancillary services such as training, waste management and demand creation were procured from manufacturers, and delivered through their networks of in-country importers and distributors. The project reportedly led to sales of 1.3–1.6 million RDTs, initiated regulatory change in 3 countries and updated policies on RDT use in all countries, and was argued in the evaluation to have fostered sustainable RDT markets in Kenya, Uganda and Tanzania, though not in Nigeria or Madagascar [67]. The UNITAID-funded team also drew on lessons from their project to produce a "Roadmap for optimizing private sector malaria rapid diagnostic testing" in collaboration with WHO, providing detailed guidance on policy implementation [69]. However, despite numerous data collection activities being undertaken, no evaluation meeting the inclusion criteria for this review was available, partly reflecting a focus on "learning by doing" during implementation, and partly the curtailment of funding before full scale-up. Other sizeable PMR projects without an evaluation eligible for this review include Defeat Malaria in DRC which involved RDT provision through pharmacies; and MalariaCare's introduction of iCCM in drug shops in Nigeria [16]; while for the similar EQuiPP project in Nigeria, the evaluation is only available in summary form with insufficient methodological detail for inclusion in this review [70].

Turning to the broader PMR interventions that covered a wider range of health conditions, the evidence base of eligible studies was very limited, and no evaluations were identified of

such interventions involving RDTs. The ADDO programme has been implemented at national scale in Tanzania, with similar projects initiated in Ghana, Uganda and Liberia [71]. However, only two studies were identified with a baseline or control, both in Tanzania, providing some evidence of increased ACT uptake. The Living Goods model of establishing CHWs as retailers did not improve ACT uptake, but did improve the quality of ACTs in the wider market. However, replicability has not been established in other settings, and evidence was not available on more recent versions of the Living Goods model implemented at large scale in Uganda and Kenya.

Only one of the evaluations eligible for this review employed any digital or online approaches for training, supervision, monitoring or surveillance of PMRs (Dieci et al. used an existing digital sales and inventory management platform to deliver patient subsidies and provider performance incentives [48]). However, multiple digital tools are being developed, tested and implemented by NGOs and private tech companies in Africa, both specific to malaria and with broader remits [72, 73]. These can be classified into three broad groups:

- PMR-facing–typically mobile app based, these include: (i) Mobile reporting of surveillance data from PMRs e.g. Private Sector Integrated Surveillance System (ISS) piloted in ADDOs, and Uganda's mTrac tool [74], which both link to DHIS2; (ii) mobile-based detailing and training programmes e.g. PSI COVID training via WhatsApp; (iii) Electronic point-of-sale (e-POS) systems, which provide a platform for PMRs to monitor their stock and support re-ordering, and can also be used to monitor their practices in real-time, offer feedback, provide bulk discounts, deliver training messages, and target subsidies e.g. MaishaMeds (as used by Dieci et al. [48]), Bloom, Shelf Life; (iv) digital reading of RDTs that automates reporting of test results e.g. Audere's HealthPulse; and (v) clinical decision-support tools e.g. THINKMD's Medsinc, SHOPSPlus TB STARR, D-Tree International's Afya-Tek referral platform.

- Patient-facing–(i) online platforms or apps for generic or targeted health messaging e.g. Viamo, askNivi; (ii) apps or card systems using mobile money to target subsidies to specific users or those testing positive e.g. mPharma's Mutti card; (iii) digital platforms linked with insurance e.g. Babyl.

- Regulator/supervisor-facing—tablet-based tools that facilitate in-person supervision or inspection e.g. PSI's Health Network Quality Improvement System (HNQIS).

Such digital tools are likely to be used in most retail interventions going forward, so the lack of evidence on their impact presents a challenge for policy makers, though it is anticipated that more data will be available in the near future. For example, the TESTsmART RCT in Kenya and Nigeria is evaluating conditional ACT subsidies and bonuses to PMR-staff delivered through a mobile app [75].

Another challenge in making policy decisions on the evidence base in this review is that ACT and RDT markets have changed considerably since the early phase of the review period, and this could have a substantial impact on intervention effects. Of the interventions involving ACT subsidies with no RDTs, all the sub-national studies began pre-2010, and the baselines for the national AMFm/CPM studies were all pre-2012. These interventions started from a low level of retail ACT sales, and quite high ACT retail prices. For example, in the AMFm countries, the baseline market share of quality-assured ACTs in private-for profit outlets was under 7% in all but one country [12], but in 2018/19 it was 21% in Kenya, 30% in Nigeria and 50% in Uganda [76]. One might therefore expect a quality-assured ACT subsidy programme introduced now to have smaller effects on ACT use than one introduced a decade ago. However, evidence on the growing predominance of non quality-assured ACTs (i.e. not meeting

international quality requirements) in retail markets in recent years [77–79], indicates that subsidies could still play a role in increasing the market share of quality-assured medicines. However, this may face resistance from those who fear this would crowd-out local manufacturers who lack the capacity to apply for pre-qualified status [18].

In addition to these developments in ACT markets, the policy environment has also changed over the course of the review period, reflecting WHO's advocacy that all suspected malaria cases receive parasitological confirmation before treatment [80]. This means that programmes to subsidise ACT in PMRs are unlikely to be supported without RDT provision. The RDT market in PMRs has also expanded over the review period, though to a much smaller degree than for ACTs. While there has been some increase in in RDT availability, the percentage of PMRs stocking RDTs has remained well below 20% [6, 79], meaning that there is still considerable growth potential. It is likely that users are far more familiar with RDTs than they were in the early period of this review, reflecting their use across the public sector in recent years, which might increase user responsiveness to interventions to enhance RDT provision in PMRs. On the other hand, all but one of the RDT and iCCM interventions included in the review were implemented in the context of subsidized ACTs, and the impact on RDT uptake and dispensing by test result in the absence of ACT subsidies is unclear.

In sum, important knowledge gaps remain, and an ongoing programme of evidence generation is needed to inform PMR interventions. We recommend that the research and policy agenda incorporate four key priorities:

First, evidence is needed on the impact of large-scale interventions to enhance the use of RDTs in guiding ACT use, covering multiple districts in a range of urban/ rural, francophone/ anglophone, and endemicity settings. Given that costs of ACT subsidies were US$336 million for AMFm Phase 1 alone [12], it is likely that strategies using much lower commodity subsidies may be required. It is also important to recognise that approaches used in earlier studies such as face-to-face training and supervision of all enrolled PMRs, may not be feasible given the sheer number of such outlets and competing demands on health sector and regulatory staff. Rather these studies should test scalable and affordable models, incorporating "smart" approaches that benefit from substantial economies of scale, such as PMR-facing digital technologies for training, supervision, monitoring, stock control and surveillance; use of mass media; and engagement with trade associations, importers and distributors. While these strategies are unlikely to be amenable to RCT designs, other robust approaches to evaluation would be possible, and it is crucial that funders of such interventions also support the generation and dissemination of this evidence, including the benefits of including the high proportion of malaria cases seen at PMRs in national surveillance systems. Implementing these strategies will require changes in regulation at least for some countries and PMR types. In 2019 of 7 countries surveyed (Chad, DRC, Ghana, Kenya, Nigeria, Tanzania, Uganda) RDTs could be administered in pharmacies in only 4 countries (DRC, Ghana, Nigeria, Uganda), while in registered drug stores in only 3 (Ghana, Nigeria, Uganda) [7]. Although it is recognised that illicit sales are common in some settings where they are not officially permitted [7], the lack of regulatory approval presents challenges for any intervention. However, expansion of point-of-care tests in PMRs for other conditions, including COVID-19, HIV and diabetes, may serve to shift the policy discourse. Similar considerations are relevant for evaluation of scale up of iCCM, which will likely also require adoption of scalable, smart intervention components.

Secondly, to enhance sustainability and government buy-in consideration should be given to the potential to integrate malaria and iCCM PMR strategies with broader health system developments, and wider private sector engagement strategies [81]. This may involve linking with PMR programmes for other health conditions such as TB [82] or family planning [83], or engagement with wider initiatives to strengthen PMR regulation and accreditation, medicine

and diagnostic test regulation, antimicrobial stewardship, or social health insurance [19]. The limited evidence base on such broader strategies also highlights the necessity for such strategies to be tied to robust evaluation.

Thirdly, we propose a set of recommendations for future evaluations. First a standard set of core indicators should be used in all evaluations going forward. Core indicators should include ACT and RDT uptake; dispensing of both antimalarials and antibiotics by test status (positive, negative, no test); and referral according to guidelines. While we recognise that different authors may favour different precise definitions of these indicators, to enhance comparability we propose that all studies at least report the primary outcomes specified for this review (see Methods). Further, we suggest that data from household surveys be broken down by treatment source (presenting for PMRs separately); that metrics for ACTs are presented for both quality-assured and all ACTs; and that ACT and RDT availability and price are also presented where feasible, to help understand the pathways to impact. We also recommend careful consideration of the choice of data collection method; for example while use of routine paper-based PMR records may be lower cost, the Hawthorne effect may lead to substantial over-estimation of study outcomes. Where feasible, we also suggest that evaluators consider including assessment of the accuracy of RDT results performed at PMRs, given that the two studies in this review that conducted study blood slides found discrepancies that could signal significant false positive cases in PMRs [44, 47]. Given the importance of understanding equity of impact, we also recommend measurement of the socio-economic status of users, for example using simplified asset indices [84].

Fourthly, to consider the relative value for money of PMR strategies, policy makers require evidence on their predicted cost-effectiveness. However, very few empirical cost and cost-effectiveness analyses of relevant interventions in sub-Saharan Africa exist [48, 85–87]. Going forward, the need to gather additional empirical cost data should be noted, together with the potential to use cost-effectiveness models to estimate outcomes such as the cost per death averted, and explore how these vary with intervention effectiveness and context [88–90].

## Supporting information

**S1 Table. Search strategy.**
(DOCX)

**S2 Table. Quality assessment checklist.**
(DOCX)

**S3 Table. Study and intervention characteristics.**
(DOCX)

**S4 Table. Study data collection methods.**
(DOCX)

**S5 Table. Quality assessment of included studies.**
(DOCX)

**S6 Table. Heat maps.**
(DOCX)

**S1 Checklist. PRISMA checklist.**
(DOCX)

## Acknowledgments

We are grateful for the advice and review of earlier drafts from Andrea Bosman, Jane Cunningham, Peter Olumese and Nicole Dagata.

## Author Contributions

**Conceptualization:** Catherine Goodman, Sarah Tougher, Theodoor Visser.

**Formal analysis:** Catherine Goodman, Sarah Tougher, Terrissa Jing Shang.

**Funding acquisition:** Catherine Goodman.

**Methodology:** Catherine Goodman, Sarah Tougher, Terrissa Jing Shang, Theodoor Visser.

**Writing – original draft:** Catherine Goodman, Sarah Tougher.

**Writing – review & editing:** Catherine Goodman, Sarah Tougher, Terrissa Jing Shang, Theodoor Visser.

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
