## [Decision Letter · Decision Letter 0]

8 Sep 2023

PONE-D-23-15382Improving malaria case management with artemisinin-based combination therapies and malaria rapid diagnostic tests in private medicine retail outlets in sub-Saharan Africa: a systematic reviewPLOS ONE

Dear Dr. Goodman,

Thank you for submitting your manuscript to PLOS ONE. After careful consideration, we feel that it has merit but does not fully meet PLOS ONE’s publication criteria as it currently stands. Therefore, we invite you to submit a revised version of the manuscript that addresses the points raised during the review process.

We look forward to receiving your revised manuscript.

Kind regards,

Raquel Inocencio da Luz, Phd

Academic Editor

PLOS ONE

Journal Requirements:

2. Thank you for stating the following in your Competing Interests section: "No authors have competing interests"

3. We note that you have included the phrase “data not presented” in your manuscript. Unfortunately, this does not meet our data sharing requirements. PLOS does not permit references to inaccessible data. We require that authors provide all relevant data within the paper, Supporting Information files, or in an acceptable, public repository. Please add a citation to support this phrase or upload the data that corresponds with these findings to a stable repository (such as Figshare or Dryad) and provide and URLs, DOIs, or accession numbers that may be used to access these data. Or, if the data are not a core part of the research being presented in your study, we ask that you remove the phrase that refers to these data.

Reviewers' comments:

Reviewer's Responses to Questions

**Comments to the Author**

1. Is the manuscript technically sound, and do the data support the conclusions?

Reviewer #1: Partly

2. Has the statistical analysis been performed appropriately and rigorously? 

Reviewer #1: No

3. Have the authors made all data underlying the findings in their manuscript fully available?

Reviewer #1: No

4. Is the manuscript presented in an intelligible fashion and written in standard English?

Reviewer #1: Yes

5. Review Comments to the Author

Reviewer #1: Major concerns:

Why authors not used Meta-analysis for pooled estimation of the study outcomes? The final recommendation of a systematic review is qualitative while you could do it.

You combined various designs in the review. The design, methods, and quality assessment of each design are different. Interventional studies not only differ in design but often indicated the impact of a special intervention which is not usually performed in routine care.

Introduction

The authors reviewed malaria case management related to appropriate treatment (Artimisinib-based combination therapy) and availability of malaria diagnostic tests and RDTs, and the author mentioned “There are widespread concerns about the quality of malaria case management” However, I am surprised that authors did not use meta-analysis and reliable evidence at the Global level for providing the study rational, and also as relevant findings for presenting and comparing in the discussion:

Availability of malaria diagnostic tests, anti-malarial drugs, and the correctness of treatment: a systematic review and meta-analysis. Malaria Journal 2023, 22(1).

Health workers readiness and practice in malaria case detection and appropriate treatment: a meta-analysis and meta-regression. Malaria Journal 2021, 20(1).

Authors also expressed that “As no comprehensive, up-to date systematic review was identified on strategies to improve care of febrile patients by PMR, WHO commissioned this review”. In addition to the above paper please see the below article: Development and validation of an online tool for assessment of health care providers’ management of suspected malaria in an area, where transmission has been interrupted. Malar J 21, 304 (2022). I think that “no comprehensive” should be replace by “limited evidence”

Methods/search strategy

Why you had no search on Scopus and ISI?

Please mention exact PubMed used search strategy in the text.

Where did you search unpublished records?

6. PLOS authors have the option to publish the peer review history of their article (what does this mean?). If published, this will include your full peer review and any attached files.

Reviewer #1: **Yes: **Hosein Azizi

---

## [Author Response · Author response to Decision Letter 0]

6 Oct 2023

Please see uploaded Response to Reviewers document, which contains our responses to the comments by both the Reviewer and the Editor.

---

## [Decision Letter · Decision Letter 1]

29 May 2024

PONE-D-23-15382R1Improving malaria case management with artemisinin-based combination therapies and malaria rapid diagnostic tests in private medicine retail outlets in sub-Saharan Africa: a systematic reviewPLOS ONE

Dear Dr. Goodman,

Thank you for submitting your manuscript to PLOS ONE. After careful consideration, we feel that it has merit but does not fully meet PLOS ONE’s publication criteria as it currently stands. Therefore, we invite you to submit a revised version of the manuscript that addresses the points raised during the review process. Apologies for the long delay, there was a problem with the revision of one reviewer and it has been challenging to secure new reviewers. I'm pleased to share the current descion, please proceed to minor revision of the manuscript

Please submit your revised manuscript by Jul 13 2024 11:59PM. If you will need more time than this to complete your revisions, please reply to this message or contact the journal office at plosone@plos.org. Please include the following items when submitting your revised manuscript:A rebuttal letter that responds to each point raised by the academic editor and reviewer(s). You should upload this letter as a separate file labeled 'Response to Reviewers'.A marked-up copy of your manuscript that highlights changes made to the original version. You should upload this as a separate file labeled 'Revised Manuscript with Track Changes'.An unmarked version of your revised paper without tracked changes. You should upload this as a separate file labeled 'Manuscript'.If applicable, we recommend that you deposit your laboratory protocols in protocols.io to enhance the reproducibility of your results. Protocols.io assigns your protocol its own identifier (DOI) so that it can be cited independently in the future. For instructions see: https://journals.plos.org/plosone/s/submission-guidelines#loc-laboratory-protocols. Additionally, PLOS ONE offers an option for publishing peer-reviewed Lab Protocol articles, which describe protocols hosted on protocols.io. Read more information on sharing protocols at https://plos.org/protocols?utm_medium=editorial-email&utm_source=authorletters&utm_campaign=protocols.

We look forward to receiving your revised manuscript.

Kind regards,

Raquel Inocencio da Luz, Phd

Academic Editor

PLOS ONE

Journal Requirements:

Additional Editor Comments:

Reviewers' comments:

Reviewer's Responses to Questions

**Comments to the Author**

1. If the authors have adequately addressed your comments raised in a previous round of review and you feel that this manuscript is now acceptable for publication, you may indicate that here to bypass the “Comments to the Author” section, enter your conflict of interest statement in the “Confidential to Editor” section, and submit your "Accept" recommendation.

Reviewer #1: All comments have been addressed

Reviewer #2: (No Response)

2. Is the manuscript technically sound, and do the data support the conclusions?

Reviewer #1: Partly

Reviewer #2: Yes

3. Has the statistical analysis been performed appropriately and rigorously? 

Reviewer #1: No

Reviewer #2: Yes

4. Have the authors made all data underlying the findings in their manuscript fully available?

Reviewer #1: No

Reviewer #2: No

5. Is the manuscript presented in an intelligible fashion and written in standard English?

Reviewer #1: Yes

Reviewer #2: Yes

6. Review Comments to the Author

Reviewer #2: My comments regarding a manuscript titled

“Improving malaria case management with artemisinin-based combination therapies and malaria rapid diagnostic tests in private medicine retail outlets in sub-Saharan Africa: a systematic review”

1. Given that an update version of PRISMA 2020 is available why you prefer to use the old one PRISMA 2015?

2. Since your review focused on Sub-Saharan Africa but you did not include African journal online in your database search, why?

3. What are the tools you used to resolve the possible existence of biases in this review?

4. I think it is better to use a mix of figures and tables to present your results.

Overall, except the comments raised above the manuscript was well written.

7. PLOS authors have the option to publish the peer review history of their article (what does this mean?). If published, this will include your full peer review and any attached files.

Reviewer #1: No

Reviewer #2: No

---

## [Author Response · Author response to Decision Letter 1]

10 Jun 2024

We have uploaded the response to reviewer's comments as a word document

---

## [Editor Report · Decision Letter 2]

12 Jul 2024

Improving malaria case management with artemisinin-based combination therapies and malaria rapid diagnostic tests in private medicine retail outlets in sub-Saharan Africa: a systematic review

PONE-D-23-15382R2

Dear Authors,

We’re pleased to inform you that your manuscript has been judged scientifically suitable for publication and will be formally accepted for publication once it meets all outstanding technical requirements.

Kind regards,

Raquel Inocencio da Luz, Phd

Academic Editor

PLOS ONE

---

## [Editor Report · Acceptance letter]

19 Jul 2024

PONE-D-23-15382R2 

PLOS ONE

Dear Dr. Goodman, 

I'm pleased to inform you that your manuscript has been deemed suitable for publication in PLOS ONE. Congratulations! Your manuscript is now being handed over to our production team.

Kind regards, 

on behalf of

Dr Raquel Inocencio da Luz 

Academic Editor

PLOS ONE